# Dietary Mg Supplementation Decreases Oxidative Stress, Inflammation, and Vascular Dysfunction in an Experimental Model of Metabolic Syndrome with Renal Failure

**DOI:** 10.3390/antiox12020283

**Published:** 2023-01-27

**Authors:** Rodrigo López-Baltanás, María E. Rodríguez-Ortiz, Juan M. Díaz-Tocados, Julio M. Martinez-Moreno, Cristina Membrives, Cristian Rodelo-Haad, M. Victoria Pendón Ruiz de Mier, Mariano Rodríguez, Antonio Canalejo, Yolanda Almadén, Juan R. Muñoz-Castañeda

**Affiliations:** 1Instituto Maimónides de Investigación Biomédica de Córdoba (IMIBIC), Reina Sofia University Hospital/University of Cordoba, 14004 Córdoba, Spain; 2Redes de Investigación Cooperativa Orientadas a Resultados en Salud (RICORS), Instituto de Salud Carlos III, 28029 Madrid, Spain; 3Vascular and Renal Translational Research Group, Biomedical Research Institute of Lleida (IRBLleida), Arnau de Vilanova University Hospital, 25198 Lleida, Spain; 4Instituto Maimonides de Investigacion Biomédica de Córdoba (IMIBIC), Unidad de Gestión Clinica Nefrología, Reina Sofia University Hospital/University of Cordoba, 14004 Córdoba, Spain; 5Department of Integrated Sciences/Research Center on Natural Resources, Health and Environment (RENSMA), University of Huelva, 21007 Huelva, Spain

**Keywords:** magnesium, metabolic syndrome, chronic kidney disease, inflammation, oxidative stress, vascular dysfunction

## Abstract

Background: Metabolic syndrome (MetS) and chronic kidney disease (CKD) are commonly associated with cardiovascular disease (CVD) and in these patients Mg concentration is usually decreased. This study evaluated whether a dietary Mg supplementation might attenuate vascular dysfunction through the modulation of oxidative stress and inflammation in concurrent MetS and CKD. Methods: A rat model of MetS (Zucker strain) with CKD (5/6 nephrectomy, Nx) was used. Nephrectomized animals were fed a normal 0.1%Mg (MetS+Nx+Mg0.1%) or a supplemented 0.6%Mg (MetS+Nx+Mg0.6%) diet; Sham-operated rats with MetS receiving 0.1%Mg were used as controls. Results: As compared to controls, the MetS+Nx-Mg0.1% group showed a significant increase in oxidative stress and inflammation biomarkers (lipid peroxidation and aortic interleukin-1b and -6 expression) and Endothelin-1 levels, a decrease in nitric oxide and a worsening in uremia and MetS associated pathology as hypertension, and abnormal glucose and lipid profile. Moreover, proteomic evaluation revealed changes mainly related to lipid metabolism and CVD markers. By contrast, in the MetS+Nx+Mg0.6% group, these parameters remained largely similar to controls. Conclusion: In concurrent MetS and CKD, dietary Mg supplementation reduced inflammation and oxidative stress and improved vascular function.

## 1. Introduction

Pathologies of high prevalence such as metabolic syndrome (MetS) and chronic kidney disease (CKD) are associated with cardiovascular disease (CVD). In both MetS and CKD, vascular alterations are causally associated with the induction of oxidative stress and inflammation.

MetS includes hypertension, central obesity, hyperglycemia, and dyslipidemia, so that itself is being recognized as an important risk factor of CDV in itself [1,2,3,4]. However, it is difficult to separate the contribution of each individual component on the development of CVD. Obesity and impaired glucose and lipid metabolism favor the release of cytokines from adipose tissue, leading to an inflammatory state [5,6] with an increase of reactive oxygen species (ROS) [7]. Both systemic and local inflammation and oxidative stress are key contributors to vascular dysfunction and cardiovascular derangements [8,9]; which, in turn, worsen the MetS [10]. 

Chronic systemic inflammation and oxidative stress are also common features of CKD [8,11,12,13] that have also been associated with CVD progression, the major cause of morbidity and mortality in CKD patients [14]. Furthermore, MetS has been clearly associated with CKD [15,16,17]. MetS can lead to changes in renal structure and function [17,18]. Nearly each component of MetS has been associated with both CKD incidence and progression, including insulin resistance, obesity, hypertension, dyslipidemia, with an outstanding role of inflammation, oxidative stress, and endothelial dysfunction (ED) [16,17,19]. Furthermore, in patients with concurrent MetS and CKD the risk of CVD events and mortality may be additive [16]. This has led to the search for effective preventive and therapeutic strategies, many of which have anti-inflammatory and antioxidative effects [16,17]. 

Interestingly, serum Mg levels have been inversely associated with both oxidative stress and inflammation. Experimental animals with Mg deficiency showed systemic inflammation [20] and oxidative stress [21]. Conversely, the administration of Mg reduced inflammatory markers [22,23] and the risk of CVD in patients with low intracellular Mg [24,25,26]. In fact, medical associations have included recommendations on Mg intake to prevent CVD [27]. Moreover, dietary Mg intake is negatively correlated with MetS [28,29] and low serum Mg is associated with CKD progression [30,31]. In the context of concurrent MetS and CKD, inflammation and oxidative stress are key factors that favor the progression of vascular damage; however, the effects of Mg supplementation have not been explored.

The aim of the present study was to evaluate whether dietary Mg supplementation may attenuate vascular dysfunction through the modulation of oxidative stress and inflammation in an experimental model of concurrent MetS and CKD (5/6 nephrectomized Zucker rats).

## 2. Materials and Methods

### 2.1. In Vivo Experiments

#### Animals and Surgical Procedures

In this study, the strain of Zucker rats with the MetS phenotype was used. Rats were aged 9–10 weeks with a weight of 250–300 g at the start of the experiments. Animals were purchased from Harlan Laboratories (Barcelona, Spain), maintained in a 12:12 h light–dark cycle and given ad libitum access to water and food. Prior to the experimental procedures, animals were fed a standard diet for 2 weeks. Thereafter, as in previous work [32], uremia was induced by 5/6 nephrectomy (Nx), a two-step procedure that reduces the original renal mass by five-sixths. After the second surgery, dietary phosphate content was changed to 0.9%, and rats were also treated with calcitriol (60 ng/kg ip, Calcijex, Abbot, Madrid, Spain) to control secondary hyperparathyroidism. The standard diet used in our studies was a semi-purified diet based on the Altromin C1031 diet (Altromin, GmbH, Lage, Germany) prepared to contain Ca 0.6%, P 0.6% and Mg 0.1%. The excess of phosphate in the diet was achieved adding inorganic salts of phosphate (sodium and potassium phosphate). For the dietary Mg supplementation, a special diet containing Ca 0.6%, P 0.9% and Mg 0.6% was used, in which the supplement of Mg was added as magnesium carbonate.

A subset of 10 rats, that served as controls, underwent a sham-surgery without nephrectomy and with MetS. All Nx rats were randomized in two groups with a different content of magnesium in the diet, normal (0.1%) or supplemented (0.6%). Therefore, the three studied groups of rats were: (1) Zucker rats with normal renal function and a MetS phenotype (MetS, n = 10); (2) Zucker rats with Nx fed a 0.6% Mg diet (MetS+Nx+Mg0.1%, n = 10) and (3) Zucker rats with Nx fed a 0.6% Mg diet (MetS+Nx+Mg0.6%, n = 10). After 4 weeks of treatment animals were sacrificed (24 h after animals received the last dose of calcitriol) and plasma, urine, kidneys, and thoracic aortas were collected. Twenty four-hour urine was collected in metabolic cages for measurement of phosphate and magnesium amount. Urine phosphate and magnesium were measured by spectrophotometry (Bio-Systems, Barcelona, Spain). All experimental protocols were reviewed and approved by the Ethics Committee for Animal Research of the University of Cordoba (Cordoba, Spain) and all rats received humane care in compliance with the National Society for Medical Research and conform to Directive 2010/63/EU.

### 2.2. Blood Pressure (BP) Measurements

Systolic and diastolic BP were measured using a noninvasive tail cuff system (LE 5001 Pressure Meter, Harvard Apparatus, Panlab, Barcelona, Spain) in all experimental groups after the 28-day period of treatment. A total of 8–10 measurements were obtained for each animal.

### 2.3. Blood Chemistry

Blood samples were obtained by abdominal aorta punction. Blood was collected in heparinized syringes and plasma was separated by centrifugation and stored at −80 °C until assayed.

Plasma creatinine, phosphate, magnesium, glucose, total cholesterol, high-density lipoprotein (HDL)-cholesterol and urine phosphate and magnesium were measured by spectrophotometry (Bio-Systems, Barcelona, Spain). No-HDL cholesterol values were obtained subtracting the values of total cholesterol the values of HDL.

ELISA tests were used to determine plasma levels of Endothelin-1 (R&D Systems, Minneapolis, MN, USA), Fibroblast Growth Factor 23 (FGF23) (Kainos Laboratories, Tokyo, Japan) and IL-6 (Cusabio, Wuhan, China). Plasma nitric oxide (NO) was quantified by the commercial kit QuantiChrom (BioAssay Systems, Hayward, CA, USA) which is designed to accurately measure NO production following reduction of nitrate to nitrite using the Griess method and reading its absorbance at 540 nm.

### 2.4. Evaluation of Oxidative Stress

Oxidative stress was assessed by measuring the activity of glutathione peroxidase (GPx) and lipid peroxides (malonyl dialdehyde, MDA) in plasma. For the measurement of GPx activity, a GPX assay kit from Cayman Chemical, Ann Arbor, MI was used. Sample availability allowed to evaluate GPx activity only in 5 Control-MetS animals, 9 MetS+Nx+Mg0.1% and 6 MetS+Nx+Mg0.6%. Lipid peroxides were measured using a lipid peroxidation assay MAK085 from Sigma-Aldrich, St. Louis, MO, USA. Both parameters are expressed according to the amount of plasma protein concentration measured by the Bradford method (Bio-Rad Laboratories, Munich, Germany).

### 2.5. Real-Time PCR

Total RNA was isolated from thoracic aorta samples using 1 mL Tri-Reagent (Sigma-Aldrich) and it was quantified by spectrophotometry (ND-1000; Nanodrop Technologies, Wilmington, DE, USA). IL-1 and IL-6 mRNA levels in thoracic aorta were measured by real-time quantitative RT-PCR (qRT-PCR; LC480; Roche Diagnostics, Barcelona, Spain) and the SensiFast SYBR No-ROX One-Step Kit (Bioline, London, UK) in a final volume of 10 μL from 50 ng of total RNA. The expression of target genes was normalized to the expression of glyceraldehyde-3-phosphate dehydrogenase (GAPDH). The rat primers for RT-PCR amplification were the following: IL-1 5′-AATAGCAGCTTTCGACAGTGAGGA-3′ and 5′-CCACAATGAGTGACACTGCCTTCC-3 (Gene ID: 24494), IL-6 5′-TTGGAAATGAGAAAAGAGTTGTGC-3′ and 5′-GGTAGAAACGGAACTCCAGAAGAC-3′ (Gene ID: 24498) and GAPDH 5′-AGGGGTGCCTTCTCTTGTGAC -3′ and 5′-TGGGTAGAATCATACTGGAACATGTAG-3′ (Gene ID: 24383).

### 2.6. Protein Extracts and Western Blot

Cytosolic proteins were isolated from renal samples in a lysis buffer containing 10mMHEPES, 10 mM KCl, 0.1 mM EDTA, 0.1 mM EGTA, 1 mM DTT, 0.5 mM PMSF, 70 μg/mL protease inhibitor cocktail and 0.5% Igepal CA-630, pH 7.9. The suspensions were centrifuged, and the supernatants (cytosolic extracts) were stored. Nuclear extracts from thoracic aorta were obtained by incubating the pellet separated from the previous cytosolic extract in a lysis buffer containing 20 mM HEPES, 0.4 mM NaCl, 1 mM EDTA, 1 mM EGTA, 1 mM dithiothreitol, 1 mM PMSF and 46 μg/mL protease inhibitor cocktail at pH 7.9. The protein concentration was determined with the Bradford method (Bio-Rad Laboratories, Munich, Germany). For western blot, equal amounts of protein (25 mg) were electrophoresed in 4–20% sodium dodecyl sulphate–polyacrylamide gradient gel (Bio-Rad Laboratories). The proteins were subsequently transferred to a nitrocellulose membrane (Bio-Rad Laboratories). The membranes were blocked with either 5% milk or 5% bovine serum albumin (BSA) for 1 h at room temperature and then incubated with primary antibody overnight at 4 °C. The used primary antibodies in the cytosolic extracts were rat anti-Klotho (1:500, Trans Genic Inc., Kobe, Japan overnight 4 °C) and mouse anti b-actin (1:1000 from Santa Cruz Biotechnology, CA, USA) as loading control. The levels of p65-NF-kb were evaluated in nuclear extract using rabbit anti p65-NF-κb (1:500 from Santa Cruz Biotechnology, Dallas, TX, USA overnight at 4 °C and mouse anti TFIIB (1:1000, Cell Signaling, Danvers, MA, USA) as loading control.

Blots were immunolabeled using either a rabbit, rat or a mouse horseradish peroxidase-conjugated secondary antibody (D1:5000; Santa Cruz Biotechnology) and developed on autoradiographic film using the ECL Western Blotting Detection System (Amersham Biosciences, Little Chalfont, UK) in LAS 4000 (GE Healthcare Life Science, Boston, MA, USA). Specific bands were quantified by densitometric analyses (by the measurement of the integrated optical density, IOD) with ImageJ software (National Institutes of Health, Bethesda, MD, USA).

### 2.7. Proteomic Studies by DIA-SWATH in Aortic Tissue

#### 2.7.1. Sample Preparation

Firstly, the most abundant serum proteins (albumin, serotransferrin and immunoglobulins) were removed using a MARS spin cartridge Ms-3 column (Agilent, Santa Clara, CA, USA) by following manufacturer protocol. Then, the eluted proteins were precipitated by TCA-acetone precipitation, resuspended in compatible buffer (RapiGest, Waters) and digested by application of trypsin with a protocol previously described [33].

#### 2.7.2. Mass Spectrometry

For LC-MS analysis purposes, DDA and DIA-SWATH acquisition modes were used. Thus, all samples were run in random order by injecting 1 mg of total digested samples (2 mL of each sample) per run in a hybrid Q-TOF mass spectrometer (Triple TOF 5600+, Sciex, Redwood City, CA, USA) coupled online to nano-HPLC (Ekspert nLC415, Eksigent, Dublin, CA, USA). The SWATH method used consisted in a TOF MS (400–1250 m/z, 50 ms acquisition time) followed by 50 windows of variable size (400–1250 m/z, 90 ms acquisition time), with a minimum size of 5 m/z. Between samples, a standard (beta-galactosidase standard digest, ABSciex) was run with three functions: autocalibration of the instrument, column cleaning to avoid carry-over, and quality control of sensitivity and chromatography. For higher sensitivity, both DDA and SWATH runs were performed at nano-flow (300 nL/min) in a 25 cm long × 75 μm internal diameter column (Acclaim PepMap 100, Thermo Scientific, Waltham, MA, USA) using a 120 min gradient from 5% to 30% B (A: 0.1% FA in water; B: 0.1% in ACN).

#### 2.7.3. Data Analysis

The MS analysis was divided in the following steps: (i) Building of a spectral library from the peptides and proteins identified from data-dependent acquisition (DDA) and gas-phase fractionation (GF) nanoLC-MS/MS runs by using pools made from all samples. This step was performed by combining the use of Protein Pilot software (v5.0.1, Sciex) with a human SwissProt protein database fasta file (June 2019) that includes RePliCal iRT peptides (PolyQuant GmbH, Bad Abbach, Germany) and PeakView software (version 2.1, (Sciex, Redwood City, CA, USA)) with the following settings: up to 10 peptides per protein, up to 7 transitions per peptide, a peptide confidence threshold of 95% and an FDR t+hreshold of 1%. A 5 min window and a tolerance of 50 ppm were used to extract the chromatographic traces of the fragments. (ii) Analysis of each sample by data-independent acquisition (DIA-SWATH-MS) nanoLC-MS/MS runs with a variable SWATH LC-MS method as indicated above and (iii) Analysis of protein identification and quantification data for the proteins contained in the spectral library previously obtained by extraction of fragment ion chromatograms using the MS/MSALL with SWATH Acquisition MicroApp (v.2.0, Sciex) and peptide retention times calibration in all runs using spiked RePliCal iRT peptides (also present in the library).

Additionally, to be as confident as possible with identifications and quantifications, only scores above 99% and FDR below 1% were included in the analysis. Following this step, MarkerView (version 1.2.1; Sciex) was used to normalize all acquired data and a table containing differential abundance was generated for statistical analysis purposes. Proteomic pathway analysis of differentially expressed proteins was performed using Reactome.

### 2.8. Statistics

Values are expressed as means ± SEM. The difference between means for two different groups was determined by *t*-test; the difference between means for three or more groups were assessed by ANOVA. A Tukey’s test was used as a post hoc procedure. A correlation study was carried out using the Pearson test. *p* values < 0.05 were considered significant.

Principal component analysis (PCA) was performed for the first three principal components using log2 transformed normalized areas. Data acquired from proteomic analysis were normalized using a free online platform (MetaboAnalyst 5.0), which included sparse partial least squares discriminant analysis (sPLS-DA), one way-ANOVA and a heatmap. Differential proteins were screened based on Fisher’s LSD test and *p* value < 0.05. STRING was used to determine protein–protein interaction (PPI) networks. Graphpad Prism software version 8.0 was used to represent an enrichment plot from significant pathways (*p* < 0.05).

## 3. Results

### 3.1. Biochemical Parameters

Plasma biochemical parameters corresponding to the experimental groups are shown in Table 1. As expected, a clear increase in creatinine was observed in the Nx rats, however, the rise in serum creatinine levels was significantly lower in the MetS+Nx+Mg0.6% group than MetS+Nx+Mg0.1%. Similarly, Nx prompted a rise in both the phosphate and the FGF23 levels. Dietary Mg supplementation significantly decreased phosphate and FGF23 levels. As expected, serum Mg increased in the Nx rats, but it was significantly higher in the group fed a Mg0.6% than those on Mg0.1%. Dietary Mg supplementation produced an increase in magnesuria, supporting the efficiency of the diet. Furthermore, a decrease in the urine phosphate concentration was observed in rats fed a Mg0.6%.

Food consumption was individually monitored throughout the experimental period. As usual, food intake by the nephrectomized (5/6 Nx) animals was decreased as compared to the Sham-operated controls. Furthermore, between the 5/6 Nx animals, food intake was higher in the group receiving the Mg supplementation. These differences are likely related to the general health state of the animals that certainly affect to appetite. In any case, both serum Mg and urinary Mg (Table 1) reflect the total amount of Mg absorbed.

Regarding the lipid profile, as expected, control rats (Sham-MetS) showed an altered lipid profile including serum levels of total cholesterol (CHOL), high-density lipoprotein (HDL) and no-HDL cholesterol (including LDL and VLDL). In comparison with these rats, the performance of 5/6 nephrectomy (MetS+Nx+Mg0.1%) increased significantly CHOL and no-HDL CHOL without significant changes in HDL levels. As compared to this group, dietary Mg supplementation (MetS+Nx+Mg0.6%) significantly decreased serum levels of CHOL and no-HDL CHOL and increased HDL. Finally, glucose was also dramatically elevated in Nx rats fed Mg 0.1% but did not change in those receiving Mg 0.6%.

The expression of renal klotho was evaluated at the protein level by western blotting. As compared to controls, a marked reduction in klotho levels was observed in the Nx-Mg 0.1% group, whereas no decrease in klotho was observed in the Nx-Mg 0.6% rats (Figure 1A,B).

### 3.2. Dietary Mg Supplementation Reduced Inflammation and Oxidative Stress in Rats with MetS and CKD

Plasma levels of the inflammatory cytokine IL-6 were used as a marker of systemic inflammation. As compared to the Sham-operated rats, the MetS+Nx+Mg0.1% group showed a remarkable increase in plasma IL-6 levels (Figure 2A). However, this elevation was partially prevented in the MetS+Nx rats receiving Mg 0.6%. The following was to evaluate whether the vascular wall itself might participate in the production of inflammatory cytokines. As shown in Figure 2B, a strong up-regulation in the expression of IL-6, as well as IL-1B (Figure 2C), was observed in the aortic tissue of MetS+Nx rats fed Mg 0.1%, which was partially prevented in the MetS+Nx rats fed the Mg 0.6% diet. Since the expression of inflammatory cytokines is regulated through the NF-κb signaling system, we analyzed its possible involvement in our model. Western blotting analysis from nuclear extracts from aortic tissue showed that, as compared to controls, the MetS+Nx+Mg0.1% presented an increase in the nuclear content of p65-NF-κb, indicating an up-regulation of the NF-κb pathway (Figure 2D,E). However, the supplementation with Mg 0.6% in the diet prevented the nuclear translocation of p65-NF-κb.

Oxidative stress is commonly observed in CKD and MetS. Specifically, the antioxidant GPx activity has been shown to be inversely related with CV risk; and a reduction in this activity has been reported to occur in MetS. Thus, we evaluated plasma GPx activity in the study groups. As shown in Figure 3A, GPx activity was clearly decreased in Nx rats on the Mg 0.1%, in comparison with the sham-operated rats. However, this increase was largely, though not completely, prevented in Nx rats on the Mg 0.6%. Moreover, as a marker of oxidative damage, lipid peroxides were evaluated in plasma by the malonyl dialdehyde (MDA) method. As shown in Figure 3B, increased levels of MDA were observed in the MetS+Nx+Mg0.1% group compared with control, but no change was detected in the MetS+Nx+Mg0.6%. Finally, there was a significant inverse correlation between the GPx activity and the MDA levels (Figure 3C).

### 3.3. Dietary Mg Supplementation Prevented Endothelial Dysfunction in Rats with MetS and CKD

Endothelin-1 (ET-1) and nitric oxide (NO) regulate endothelial function so alterations in their normal effects leads to endothelial dysfunction with the progression of vascular disease. Figure 4A shows that the plasma levels of ET-1 in MetS+Nx rats receiving the normal Mg 0.1% were increased by 170% as compared to controls. Nonetheless, no elevation in ET-1 was observed in MetS+Nx rats receiving Mg 0.6%. In addition, NO levels were slightly, but significantly reduced in the MetS+Nx+Mg0.1% group as compared to controls (Figure 4B), while in the MetS+Nx+Mg0.6% group NO levels were not different than the controls and MetS+Nx+Mg0.1% rats.

As a comprehensive marker of endothelial function, both systolic (SBP) and diastolic blood pressure (DBP) were measured at the end the experimental period. According to previous studies from our group, normal values of blood pressure from healthy rats correspond with 125 ± 3.51 mmHg for systolic pressure and 106 ± 3.37 for the diastole [34]. As expected for the strain of Zucker rats, high values of SBP (Figure 4C) and DBP (Figure 4D) were observed in the sham-operated. Blood pressure values were significantly increased in the Nx-Mg 0.1% group. By contrast, no elevation was observed in the Nx-Mg 0.6% group, showing values similar to those of controls.

### 3.4. Dietary Mg Supplementation Altered Proteomic Markers of MetS, CKD and Vascular Dysfunction

To identify candidate serum molecules associated with the vascular derangements related to the concurrence of MetS and CKD and the protective effect exerted by the Mg supplementation, differential serum proteomic analysis was performed. A total of 195 proteins were quantified. From these, 22 proteins with a *p* value ≤ 0.05, and an FC cut-off of 2.5 were considered to be differentially abundant in the experimental groups (Table 2). Partial least squares discriminant analysis (PLSDA) revealed that the three groups clustered distinctly from one another (Figure 5A). A clear stratification according to the induction of CKD (controls vs. nephrectomized animals) was observed along the first component, which also separates largely the nephrectomized groups receiving the different dietary Mg loads. Moreover, this is also evidenced by the hierarchical clustering based in normalization relative abundance of each protein (Figure 5B).

Differential abundance levels of the individual proteins in the controls and nephrectomized animals, after being normalized with respect to the control group and represented as −log10, are shown in Figure 6. Furthermore, to gain molecular perspective on the functional significance of these proteins, they were analyzed though the Reactome database (though the proteins fetuin B, T-kininogen 1, α-2 antiplasmin and carboxylasterase 1C were not recognized by the database) (Figure 7). Pathway enrichment analysis highlighted the modulation of the following important pathways: Peptide ligand-binding receptors; detoxification of ROS; regulation of lipid metabolism by PPARalpha; plasma lipoprotein assembly, remodeling, and clearance; binding and uptake of ligands by scavenger receptors; metabolism of proteins; hemostasis; and immune system (Supplementary Material).

Among the plasma lipoprotein assembly, remodeling, and clearance proteins (Figure 6 and Appendix A), Phosphatidylcholine-sterol acyltransferase showed increased levels in both nephrectomized groups while ApoB was increased in the MetS+Nx+Mg0.1% group, but this increase was prevented by the 0.6% Mg supplementation. Moreover, ApoB was the only protein included in the binding and uptake of ligands by scavenger receptors cluster (Figure 6 and Appendix A). Identified proteins included in the pathway for regulation of lipid metabolism by PPARalpha were corticosteroid-binding globulin and AGT, which were down- and up-regulated, respectively, in the Nx+Mg0.1% group but remained with similar levels as controls in the Nx+Mg0.6% (Figure 6 and Appendix A).

Regarding the immune System pathway, seven proteins, coagulation factor XIII B chain, plasma protease C1 inhibitor, lysozyme, cathepsin-C, complement factor D, cystatin-C and apolipoprotein B100 showed an increase in abundance after nephrectomy, but this was partially and totally prevented by 0.6% Mg supplementation for the Cystatin-1 and ApoB, respectively. Moreover, down-regulation of murinoglobulin-1 induced in the Nx+Mg0.1% was not observed in the animals fed the 0.6% Mg diet (Figure 6 and Appendix A).

All seven proteins belonging to the hemostasis pathway, plasma protease C1 inhibitor, lysozyme, cathepsin-C, complement factor D, coagulation, factor XIII B chain and inter-alpha-trypsin inhibitor heavy chain H3, were up-regulated in both nephrectomized groups but the up-regulation of KIN1 was partially, and that of ApoB and ATIII totally, prevented in the group receiving the Mg supplementation. The pathway of detoxification of ROS was represented by the GPx3, which was negatively regulated after nephrectomy under both dietary Mg regimens (Figure 6 and Appendix A). We also identified several proteins that function in the cluster of metabolism of proteins, including ATIII, KIN1, KIN2, LYZ, APOB, AGT and CYSC that were positively regulated in the Nx+Mg0.1% group, but this increase was partially or totally precluded in the groups receiving the Mg supplementation, except for LYZ (Figure 6 and Appendix A). Moreover, phosphatidylinositol-glycan-specific phospholipase D showed lower abundance in both groups, while MUG1 changed as described previously. Finally, the peptide ligand-binding receptors pathway (Figure 6 and Appendix A) included two proteins, KIN1 and AGT that were modulated as stated previously.

## 4. Discussion

In the present study it was evaluated whether in a rat model of MetS (Zucker strain) with CKD, dietary Mg supplementation could improve vascular function through a decrease of oxidative stress and inflammation. Our results showed that as compared to the sham-operated rats with MetS fed a normal (0.1%) Mg, the rats with MetS and CKD fed the same Mg (0.1%) had a significant increase in oxidative stress and inflammation biomarkers as well as in blood pressure and ET-1 levels, and also a decrease in NO. By contrast, in rats with MetS and CKD receiving a dietary Mg supplementation (0.6%), the levels of these parameters remained similar to those in the sham-operated rats.

Both inflammation, and oxidative stress are hallmarks of MetS and CKD that contribute to vascular derangements such ED or atherosclerosis [6,8,10,11,12,13]. Our results showed that in rats with concurrent MetS and CKD fed a normal Mg (0.1%) diet, there was a vascular inflammatory response, as deduced from the elevation in the expression of the inflammatory cytokines IL-6, IL-1 and their up-stream regulator NF-κb in aortic tissue. This was highly blunted in the matching group receiving the Mg 0.6% supplementation. In previous in vitro studies we observed that Mg supplementation prevented the tumor necrosis factor-α-induced upregulation of the p65 and the expression of bone morphogenetic protein -2 that was observed in human umbilical vein endothelial cells incubated with a normal Mg concentration [32]; and, similarly, in VSMCs cultured in a pro-inflammatory high phosphate medium, incubation with Mg 1.6 mM inhibited the increase in the production of ROS, the rise in the expression of inflammatory cytokines and the activation of NF-κb [35]. Our findings underline that simultaneous conditions of MetS and CKD contribute to a significant inflammation of vascular wall and this pathological process appears as being modulated by dietary Mg.

Oxidative stress has been widely associated with MetS and CKD [10,11]. In our study, plasma GPx activity was inhibited in rats with MetS and CKD fed a normal Mg (0.1%) diet, which likely may result in the occurrence of oxidative stress. Certainly, the inhibition of GPx activity correlated with the increase in MDA, a biomarker of oxidative damage derived from lipid peroxidation. Conversely, both GPx activity and MDA levels remained similar to controls in the group supplemented with Mg 0.6%. The kidney is the primary organ responsible for production of GPx which is subsequently secreted into plasma; and a deficiency in this enzyme during CKD in patients and experimental animals has been related with cardiovascular events [36,37]. Of note, in MetS patients LDL peroxidation is elevated, which favors atherosclerosis and CVD in CKD [38]. Therefore, the prevention of the decrease in plasma GPx activity by Mg supplementation might likely protect from excessive oxidative stress and, therefore from the MetS- and CKD-dependent increased risk of CVD. Moreover, the high levels of oxLDL are cytotoxic in the kidney and it was observed that reduced the expression of renal klotho [39]. Therefore, it is likely that the preservation of renal klotho and the renal function by Mg supplementation that we have observed might be also mediated through the reduction of oxidative stress. The effects of Klotho inhibiting oxidative stress have been already described in other tissues and cells [40,41]. Furthermore, magnesium supplementation might have also prevented from parenchymal injury in the kidneys, as it was observed previously in the same model of CKD but without MetS [35], which might have been also facilitated by Klotho recovery.

A protective effect of Klotho on atherosclerosis and ED has been also demonstrated [42,43]. Furthermore, the amelioration of uremia by dietary Mg supplementation was associated with the total or partial correction of other parameters of mineral metabolism as phosphate and FGF23; abnormal mineral metabolism parameters have been related to the progression of cardiovascular dysfunction [44].

Additionally, since oxidative stress has been shown to be associated with alterations in both the ET-1 and NO signaling pathways [45], the modulation of oxidative stress exerted by the Mg supplementation might also have had a role in the maintenance of ET-1 and NO levels and, therefore, of blood pressure and vascular function, that we have observed.

It is important to note that dietary Mg supplementation improved the lipid profile of animals with MetS and CKD, decreasing CHOL and no-HDL CHOL and increasing HDL, which agrees with previous publications [46]. Interestingly, our proteomic profiling also revealed changes in the levels of proteins related to lipids and lipoproteins metabolism that were contrarily associated with both the nephrectomy and the beneficial effect of Mg supplementation through impacting two key reactome pathways as that of regulation of lipid metabolism by PPARalpha and that of the plasma lipoprotein assembly, remodeling, and clearance. Thus, LCAT protein, which has been inversely associated with the progression of nephropathy [47], was similarly decreased in both Nx groups. However, while AGT, ApoB and CBG were highly deregulated in the Nx-0.1% Mg rats, they remained similar to controls in rats supplemented with Mg 0.6%. Interestingly, all of them are related to CVD. It has been described as an inverse association of CBG with CVD and inflammatory risk [48]. Apolipoprotein B-100 (ApoB), an essential component of VLDLs and its metabolites and a key structural protein component of all major atherogenic lipoproteins [49], is a good indicator of CVD, insulin resistance, MetS and inflammation [50]. Angiotensinogen (AGT), a precursor of angiotensin II, is a key element of the renin angiotensin system (RAS), which is strongly implicated in CVD [51], and angiotensin II-independent effects on renal function have also been stablished [52]. Of note, these proteins appear also involved in other important pathways such as the immune system, hemostasis and metabolism of proteins. It is likely that the modulation of two pleiotropic proteins such as ApoB and AGT, with many physiological effects closely related to CVD risk, is a mechanism underlying the protective effect of dietary Mg that we have observed.

Proteomic analysis also highlighted detoxification of ROS as another important pathway through the modulation of serum abundance of the extracellular GPx3, an antioxidant enzyme that appeared downregulated in the nephrectomized animals as compared to controls. Patients with CKD show a deficiency in this enzyme [53] and it is an important factor that contributes to the development of heart disease induced by kidney failure [54]. In our study, no difference in the amount of GPx3 was observed between the two regimens of dietary Mg, although GPx activity measured in plasma was more reduced in the 0.1% Mg-rats than in those supplemented with Mg 0.6%. Moreover, the immune system and hemostasis are the pathways that include more proteins showing altered abundance as a result of the concurrence of MetS and CKD. Overall, the levels of these proteins appeared increased after nephrectomy. This includes KNG, an important pro-inflammatory and pro-oxidant factor belonging to the kallikrein-kinin-system that regulates cardiovascular and renal function [55,56]; Cystatin-C, a recognized biomarker of kidney damage, MetS and CVD predictor [57,58]; as well as the previously cited ApoB. Interestingly, however, this increase was totally or partially prevented by dietary Mg supplementation. As it was the reduction observed in the Nx-0.1% Mg group in the levels of MUG1, which has been inversely associated with the inflammatory response [59].

In addition to these vascular effects, the beneficial effects of Mg supplementation and, thus, the importance in the prevention of hypomagnesemia have been widely described by others. Serum Mg concentrations ranging from 1.73 mg/dL to 2.16 mg/dL are associated with a linear decrease in the risk of cardiovascular events; as well as a nonlinear inverse association with cardiovascular events [60]. Similarly, pathophysiological processes such as vascular calcifications, hypertension, arrhythmias, heart failure, coagulation, endothelial dysfunction, even cardiovascular death, are possibly related to abnormal Mg homeostasis [61]. Mg deficiency accelerates the atherosclerotic process, increases thromboxane synthesis and stimulates platelet aggregation, produces oxidative stress and inflammation, and induces the synthesis of cytokines, nitric oxide and adhesion molecules (VCAM-1 and ICAM-1) all of which ends in CVD [62].

Treatment of MetS is beneficial to prevent and delay the progression of kidney disease [17]. Patients with uncontrolled MetS were shown to have a higher risk of rapid decline in renal function compared to those controlled through dietary patterns [63]. In agreement with this, in our study, a dietary Mg supplementation was able to prevent the worsening of different factors of MetS, such as hypertension, glucose and lipid profile, and the development of uremia; and this was accompanied with the mitigation of inflammation and oxidative stress and the protection of the vascular function. Furthermore, all this was associated with alterations of proteomic markers of MetS, CKD and CVD towards control values. These findings support a possible beneficial effect of Mg supplementation in patients with concurrent MetS and CKD to reduce oxidative stress and inflammation and, therefore, vascular dysfunction.

There are limitations in this study that deserve consideration. We are aware that despite the fact that the 0.6% Mg supplementation of our experimental diet is acceptable for nephrectomized rats fed a high phosphate diet in the short term, this Mg supplementation cannot be directly extrapolated to the human setting. Conversely, though our results might support a possible beneficial effect of Mg supplementation, appropriate clinical trials would be desirable to stablish a safe therapeutic windows of Mg dosage, as well as to ensure in which patients, more likely those with hypomagnesemia, supplementation might be indicated.

## 5. Conclusions

In conclusion, in a rat model of concurrent MetS and CKD, dietary Mg supplementation provided protection against inflammation and oxidative stress, allowing a better control of MetS and CKD and an improvement in vascular function.

## 6. Patents

These results have been included in the national patent request with reference P201930979 and titled “Composición para la prevención, mejora, alivio y/o tratamiento de una enfermedad provocada por la inflamación y el estrés oxidativo sobre células musculares lisas vasculares”.

## Figures and Tables

**Figure 1 antioxidants-12-00283-f001:**
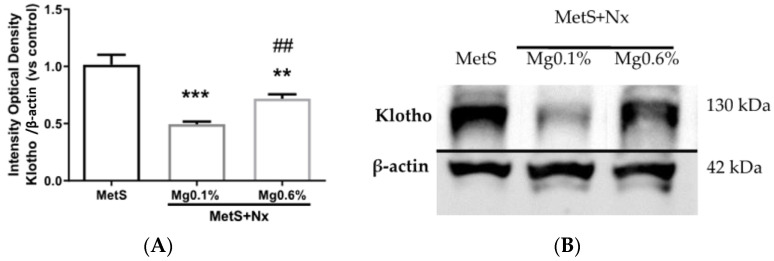
Dietary Mg supplementation prevented a decrease in renal Klotho expression in rats with metabolic syndrome and 5/6 nephrectomy. Nx rats were fed a normal (0.1%) Mg diet or received a dietary Mg supplementation (0.6%). Sham-operated rats fed a normal P and Mg diet served as controls. (**A**) Image J software was used to quantify renal Klotho expression obtained by western blotting in all animals from the study (n = 10 animals per group). (**B**) Western blot bands were cropped showing each experimental group, choosing the animal with the most approximated value to the mean of the group obtained after Image J quantification. Data expressed as mean ± SEM. *** *p* < 0.001, ** *p* < 0.01 vs. MetS and ## *p* < 0.01 vs. MetS+Nx+Mg0.1%.

**Figure 2 antioxidants-12-00283-f002:**
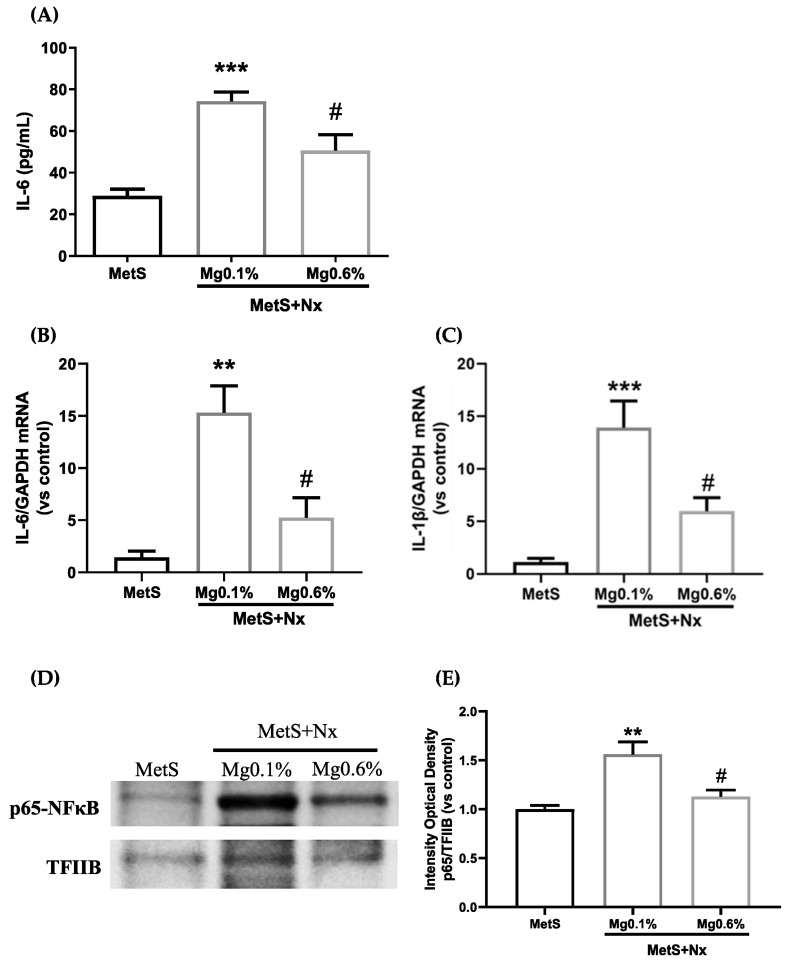
Dietary Mg supplementation reduced pro-inflammatory cytokine expression in plasma and aortic tissue induced in rats with metabolic syndrome and 5/6 nephrectomy. Nx rats were fed a normal (0.1%) Mg diet or received a dietary Mg supplementation (0.6%). Sham-operated rats fed a normal P and Mg diet served as controls. (**A**) Plasmatic levels of Interleukin-6. (**B**,**C**) mRNA levels of IL-6 and IL-1β were analyzed by real-time RT-PCR. (**D**) The p65-NF-κb (p65) protein was determined by western blotting in nuclear extracts. Each lane is representative of each experimental group, and it was included according to the most approximated value to the mean of its group obtained after Image J quantification. (**E**) Western blot quantification was performed using Image J measuring the integrated optical density and normalized to TFIIB levels. Data expressed as mean ± SEM. *** *p* < 0.001, ** *p* < 0.01 vs. MetS and # *p* < 0.05 vs. MetS+Nx+Mg0.1%.

**Figure 3 antioxidants-12-00283-f003:**
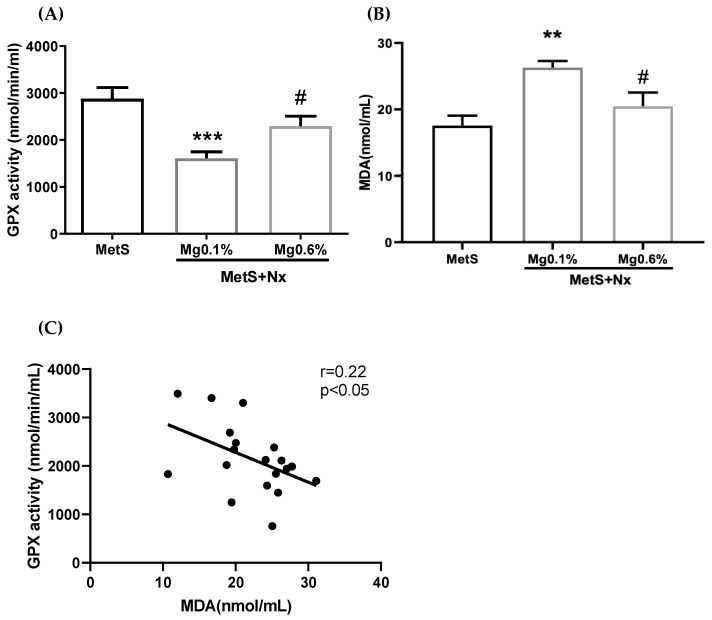
Dietary Mg supplementation reduced plasmatic oxidative stress markers in rats with metabolic syndrome and 5/6 nephrectomy. Nx rats were fed a normal (0.1%) Mg diet or received a dietary Mg supplementation (0.6%). Sham-operated rats fed a normal P and Mg diet served as controls. (**A**) Plasma GPx activity was spectrophotometrically evaluated. (**B**) Lipid peroxidation (MDA) was measured in plasma from the different experimental groups. (**C**) A negative correlation between GPx activity and MDA levels was observed. *** *p* < 0.001 and ** *p* < 0.01 vs. MetS; # *p* < 0.05 vs. MetS+Nx+Mg0.1%.

**Figure 4 antioxidants-12-00283-f004:**
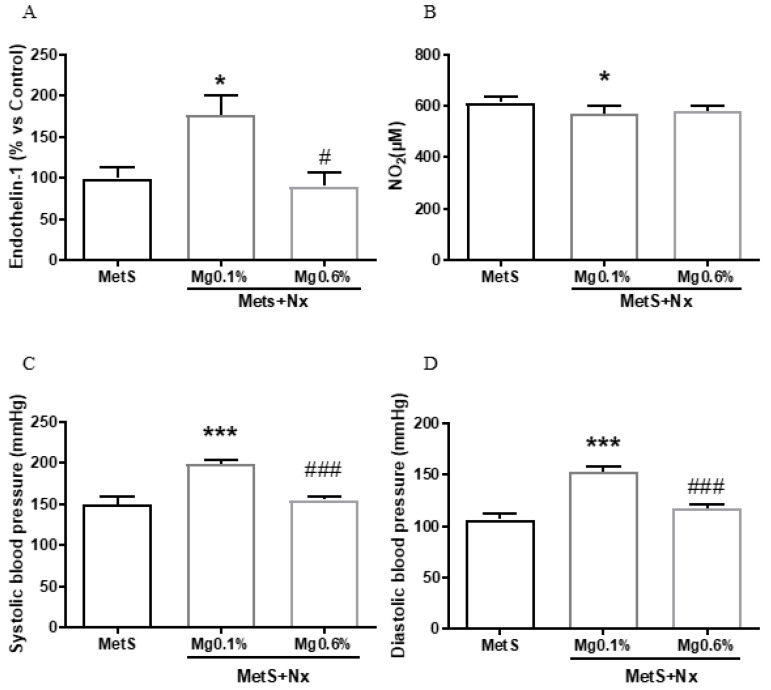
Dietary Mg supplementation improved vascular function from rats with metabolic syndrome and 5/6 nephrectomy. Nx rats were fed a normal (0.1%) Mg diet or received a dietary Mg supplementation (0.6%). Sham-operated rats fed a normal P and Mg diet served as controls. (**A**) Plasma Endothelin-1 levels. (**B**) Plasma nitric oxide (NO) levels. (**C**,**D**) Systolic and diastolic blood pressure. Data expressed as mean ± SEM. *** *p* < 0.001 and * *p* < 0.05 vs. MetS; # *p* < 0.05 and ### *p* < 0.001 vs. MetS+Nx+Mg0.1%.

**Figure 5 antioxidants-12-00283-f005:**
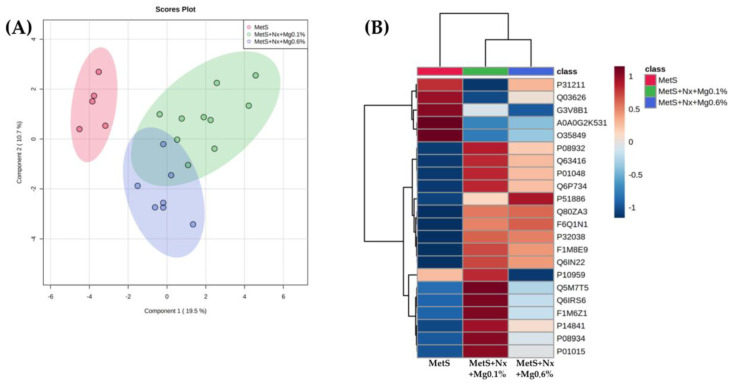
Dietary Mg supplementation modified the expression of 22 proteins related to inflammation, oxidative stress and atherosclerosis in rats with metabolic syndrome and 5/6 nephrectomy. A Plasma proteomic study was performed by DIA/SWATH. (**A**) Principal component analysis (PCA) score plot from fold-change values of the different experimental groups. (**B**) Heat map analysis of the 22 proteins with significantly altered expression levels. All the significantly altered proteins were grouped into 3 clusters (Red = MetS; Green = Mets+Nx+Mg0.1% and Blue = Mets+Nx+Mg0.6%).

**Figure 6 antioxidants-12-00283-f006:**
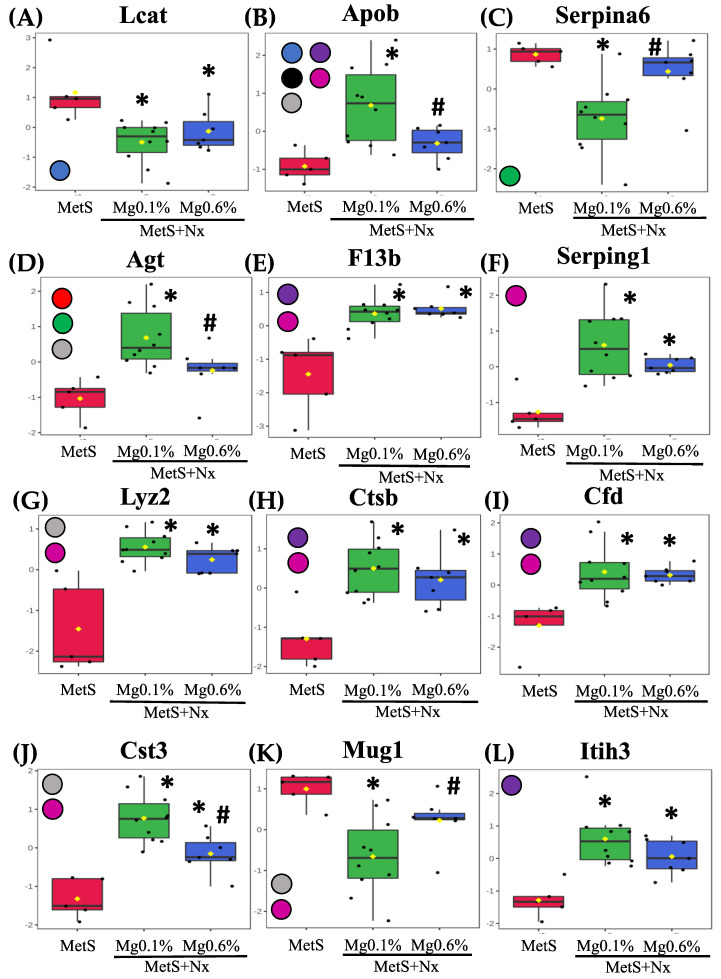
Plasma proteins differentially expressed in the proteomic study. After plasma proteomic study by DIA-SWATH the following proteins were statistically significant after comparison between the experimental groups: (box plots are labeled with gene names that encode the respective proteins). (**A**) Phosphatidylcholine-sterol acyltransferase (Lcat), (**B**) Apolipoprotein B-100 (ApoB), (**C**) Corticosteroid-binding globulin (Serpina6), (**D**) Angiotensinogen (Agt), (**E**) Coagulation factor XIII B chain (F13b), (**F**) Plasma protease C1 inhibitor (Serping1), (**G**) Lysozyme (Lyz2), (**H**) Cathepsin B (Ctsb), (**I**) Complement factor D (Cfd), (**J**) Cystatin-C (Cst3), (**K**) Murinoglobulin-1 (Mug1), (**L**) Inter-alpha-trypsin inhibitor heavy chain H3 (Itih3), (**M**) Kininogen-1 (Kng1), (**N**) Antithrombin-III (Serpinc1), (**O**) Glutathione Peroxidase 3 (Gpx3) (**P**) Kininogen-2 (Kng2), (**Q**) Phosphatidylinositol-glycan-specific phospholipase (Gpld1), (**R**) Fetuin B (Fetub), (**S**) T-kininogen 1 (Map1), (**T**) Alpha2-antiplasmin (Serpinf1), (**U**) Carboxylesterase 1C (Ces1c) and (**V**) Lumican (Lum). Intensities were normalized to the mean value of each protein. The participation of these proteins in the different biochemical pathways according to reactome analyses is indicated through color circles in the left margin from each graphic. Red circle: Peptide ligand-binding receptors; Orange circle: Detoxification of ROS; Green circle: Regulation of lipid metabolism by PPARalpha; Blue circle: Plasma lipoprotein assembly, remodeling, and clearance; Black circle: Binding and uptake of ligands by scavenger receptors; Grease circle: Metabolism of proteins; Violet circle: Hemostasis and Pink circle: Immune system. * *p* < 0.05 vs. MetS group and # *p* < 0.05 vs. MetS+Nx+Mg0.1% group.

**Figure 7 antioxidants-12-00283-f007:**
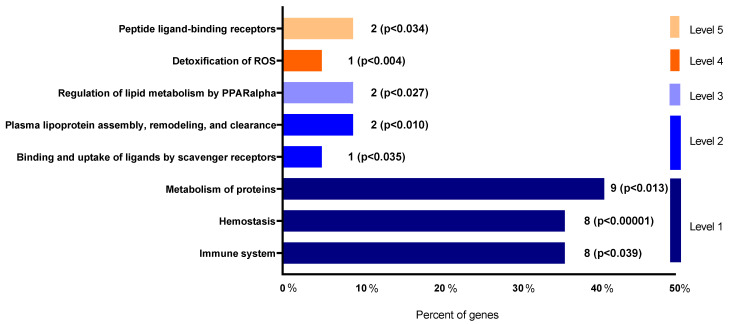
Functional significance of the identified proteins on biochemical pathways according to Reactome database.

**Table 1 antioxidants-12-00283-t001:** Serum and urine biochemical parameters.

	Sham -MetS	MetS+Nx +Mg0.1%	MetS+Nx +Mg0.6%
Creatinine (mg/dL)	0.52 ± 0.1	1.59 ± 0.2 *	0.99 ± 0.1 *#
Phosphate (mg/dL)	8.75 ± 0.35	13.05 ± 0.9 *	10.63 ± 0.4 *#
Magnesium (mg/dL)	2.30 ± 0.2	3.97 ± 0.3 *	6.24 ± 0.8 *#
Glucose (mg/dL)	269 ± 54.5	426.7 ± 34.5 *	297.7 ± 35.1
FGF23 (pg/mL)	692.5 ± 32.2	2275 ± 270.1 ***	1229 ± 142.1 ##
Total cholesterol (mg/dL) *	256.2 ± 19.5	326 ± 13.5 **	285.3 ± 19.6#
HDL (mg/dL) *	67.06 ± 0.6	64.05 ± 7.6	82.7 ± 5.6 *#
No-HDL cholesterol (mg/dL) *	189.1 ± 19.8	256.6 ± 14 **	202.6 ± 14.9#
Urine phosphate (mg/24 h)	2.98 ± 0.4	24,97 ± 2.3 *	12.11 ± 0.8 *#
Urine magnesium (mg/24 h)	7.08 ± 1.3	3.37 ± 0.2 *	8.02 ± 0.9 #

* *p* < 0.05, ** *p* < 0.01 and *** *p* < 0.001 vs. Sham-MetS; # *p* < 0.05 and ## *p* < 0.01 vs. MetS+Nx+Mg0.1%.

**Table 2 antioxidants-12-00283-t002:** Plasma proteins differentially expressed among the experimental groups analyzed by DIA-SWATH.

Protein Accession	Gene	Protein	F Value	*p* Value	-LOG(10)	FDR
A0A0G2K531	Gpx3	Glutathione peroxidase 3	27.743	0.00	56.366	0.0005
P14841	Cst3	Cystatin-C	22.699	0.00	50.362	0.0008
Q6IRS6	Fetub	Fetuin B	21.764	0.00	49.145	0.0008
P08932	Kng1	Kininogen-1	19.747	0.00	46.394	0.0010
F1M8E9	Lyz2	Lysozyme	19.449	0.00	45.972	0.0010
Q80ZA3	Serpinf1	Alpha-2 antiplasmin	17.566	0.00	43.197	0.0014
F6Q1N1	F13b	Coagulation factor XIII B chain	17.432	0.00	42.992	0.0014
P51886	Lum	Lumican	16.253	0.00	41.145	0.0019
P08934	Kng2	Kininogen-2	15.403	0.00	3.976	0.0023
P01048	Map1	T-kininogen 1	14.581	0.00	38.375	0.0028
Q63416	Itih3	Inter-alpha-trypsin inhibitor heavy chain H3	12.493	0.00	34.634	0.0061
Q6P734	Serping1	Plasma protease C1 inhibitor	11,91	0.00	33.525	0.0072
Q6IN22	Ctsb	Cathepsin B	10.841	0.00	31.412	0.0108
P32038	Cfd	Complement factor D	10.397	0.00	30.502	0.0124
P31211	Serpina6	Corticosteroid-binding globulin	96.864	0.00	29.001	0.0164
P01015	Agt	Angiotensinogen	93.701	0.00	28.315	0.0180
Q5M7T5	Serpinc1	Antithrombin-III	85.016	0.00	26.371	0.0265
Q03626	Mug1	Murinoglobulin-1	81.335	0.00	25.518	0.0284
P10959	Ces1c	Carboxylesterase 1C	80.811	0.00	25.396	0.0284
F1M6Z1	Apob	Apolipoprotein B-100	80.625	0.00	25.352	0.0284
O35849	Lcat	Phosphatidylcholine-sterol acyltransferase	76.767	0.00	24.435	0.0334
G3V8B1	Gpld1	Phosphatidylinositol-glycan-specific phospholipase D	70.004	0.01	22.778	0.0468

## Data Availability

All the data and results obtained during the current study are available from the corresponding author on reasonable request. Data related to SWATH-DIA study are available in the publicly archived datasets Proteomexchange.

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
