# Peer review of "Dietary Mg Supplementation Decreases Oxidative Stress, Inflammation, and Vascular Dysfunction in an Experimental Model of Metabolic Syndrome with Renal Failure"

_antioxidants, 2023, doi:10.3390/antiox12020283_

Round 1

Reviewer 1 Report

The manuscript “Dietary Mg supplementation decreases oxidative stress, inflammation, and vascular dysfunction in an experimental model of metabolic syndrome with renal failure” by López-Baltanás and colleagues reports on the effects of magnesium supplementation on several markers of inflammation, oxidative stress, endothelial function, lipid profile and blood pressure, in an animal model of metabolic syndrome with chronic kidney disease. 

The study is well-designed and its results provide interesting indications for future research on the topic.

I have some minor recommendations to improve the manuscript:

  1. More information on the composition of the diet, apart from the Mg percentage, would be useful. If a standard diet has been used, please reference it. Otherwise, provide details on how it is prepared.
  1. Was food was provided ad libitum? And if so, was the food intake of animals measured, to make sure that there was no difference in food intake (and thus of Magnesium) between the different experimental groups? 
  1. Was the average daily intake of Magnesium per body weight of the animals estimated? 
  1. It would be very useful to make an estimation of how the Mg content in the diet would translate to a human diet. For example, you could use a body surface area method to extrapolate to how many milligrams of Mg per body weight, the dose you gave would correspond to humans, or at least, to how many milligrams of Mg per day the dose you gave would correspond in an average human diet.
  1. In the abstract, you mention that control rats have undergone a sham operation, however there is no mention of this procedure in section 2.1.1 on surgical procedures, from which it would appear as if control rats did not undergo any surgical procedure at all. You may add a sentence between lines 90 and 91, stating something like “A subset of 10 rats, that served as controls, underwent a sham-surgery without actual nephrectomy”.
  1. Also, there is no mention in the methods section about urine collection and what analyses were performed in urines and using what methods
  1. In Line 209, in the Statistics section, you say that values are expressed as means±SE. However, when results are reported (eg in Figures 1, 2 and 4) the legend says that data are means±SEM. One of the two must be wrong.
  1. In Table 1, at the line for Glucose, it would appear as if there is no statistically significant difference between the Mg0.1 and Mg0.6 groups, but there clearly is a major difference, so probably there’s a missing “###” sign.
  1. From your discussion, it would appear that the only mechanism you propose to explain the effects of Mg is its modulation of ApoB and AGT. But this is extremely reductive, considering that Mg is involved in over 300 enzymatic pathways, and there are a lot of other mechanisms that could explain its beneficial effects (for example it involvement in arterial muscle vasoconstriction, vasomotor tone, blood pressure, blood coagulation, insulin metabolism…). I would recommend to add at least a sentence about the possible mechanisms of action of Mg, and maybe reference one or two reviews that explain the many different mechanisms through which Mg may be involved in relation to CVD protection.
  1. Line 113, how could you use the Friedewald formula if you did not measure triglycerides
  1. Following are some minor misspells and typos:

Lines 29-30, spell out IL-1, IL-6, ET-1 and NO 

Line 35, replace “as well as improvement in” with “, and improved”

Line 45, replace “itself is being recognized as an important risk factor of CDV” with “is being recognized as an important risk factor of CDV in itself”

Line 46, replace “components” with “component”

Line 51, replace “worsens” with “worsen”

Line 55, replace “chronic kidney disease” with “CKD”

Line 66, delete “Mg”

Line 69, replace “with” with “is”

Line 69, replace “progression CKD” with “progression”

Line 80, replace “the metabolic syndrome (MetS)” with “MetS”

Line 82, open parenthesis before Barcelona, Spain

Line 82, replace “and” with “,”

Line 92, replace “metabolic syndrome” with “MetS”

Line 107, I would hope samples were stored at “–80°C”, not 80°C!

Line 115, spell out FGF23

Line 116, abbreviate nitric oxide with NO

Line 156, add beta before -actin

Line 210, “with statistical.”: something is missing, probably the name of the statistical software.

Line 212, merge sentence with line 211

Line 213, replace “was considered” with “were considered”

Line 215, spell out PCA

Line 278, delete “it was observed”

Line 292, in subfigures D and E please clarify on what tissue was the analysis performed (eg. “Nuclear extracts from aortic tissue”)

Line 339, delete the second “A)”

Line 360, replace “normalized” with “normalization”

Line 441, replace “improve” with “could improve”

Line 444, replace “there was” with “had”

Lines 455-457, spell out the acronyms

Line 468, replace “to produce GPx” with “for production of GPx”

Line 468, replace “excreted” with “secreted”

Line 504, spell out RAS

Line 506, replace “has” with “have”

Line 511, replace “other” with “another”

Line 512, replace “though” with “through”

Line 519, replace “includes” with “include”

Author Response

POINT-BY-POINT RESPONSE TO THE REVIEWERS COMMENTS

REVIEWER 1

Undoubtedly, the manuscript has improved considerably after reviewer 1 comments and questions. We are thankful for this kind and constructive revision.

  1. More information on the composition of the diet, apart from the Mg percentage, would be useful. If a standard diet has been used, please reference it. Otherwise, provide details on how it is prepared.

We greatly appreciate the suggestion made by the reviewer and recognize that we have provided little information about this issue. This information has been added to the revised version of the manuscript in the Materials and Methods section: Page 2, Line 88.

“The standard diet used in our studies was a semi-purified diet based on the Altromin C1031 diet (Altromin, GmbH, Lage, Germany) prepared to contain Ca 0.6%, P 0.6% and Mg 0.1%. The excess of Phosphate in the diet was achieved adding inorganic salts of phosphate (sodium and potassium phosphate). For the dietary Mg supplementation, a special diet containing Ca 0.6%, P 0.9% and Mg 0.6% was used, in which the supplement of Mg was added as Magnesium carbonate”.

  1. Was food was provided ad libitum? And if so, was the food intake of animals measured, to make sure that there was no difference in food intake (and thus of Magnesium) between the different experimental groups? 

Food was provided ad libitum, but food consumption was individually monitored throughout the experimental period. As usual, food intake by the nephrectomized (5/6 Nx) animals was decreased as compared to the Sham-operated controls. Furthermore, between the 5/6 Nx animals, food intake was higher in the group receiving the Mg supplementation. These differences are likely related to the general health state of the animals that certainly affect to appetite. In any case, both, serum Mg and urinary Mg (Table 1) reflects the total amount of Mg absorbed.

First line of results section:

“Food consumption was individually monitored throughout the experimental period. As usual, food intake by the nephrectomized (5/6 Nx) animals was decreased as compared to the Sham-operated controls. Furthermore, between the 5/6 Nx animals, food intake was higher in the group receiving the Mg supplementation. These differences are likely related to the general health state of the animals that certainly affect to appetite. In any case, both, serum Mg and urinary Mg (Table 1) reflects the total amount of Mg absorbed”.

  1. Was the average daily intake of Magnesium per body weight of the animals estimated? 

At the beginning of the experiments all animals weighted approximately the same (mean: 277 g), and the mean daily intake of food was 12 g. Therefore, the mean daily intake of Magnesium per body weight was 43.6 mg/kg for animals receiving Mg 0.1% and a six fold for those supplemented with Mg 0.6%.

  1. It would be very useful to make an estimation of how the Mg content in the diet would translate to a human diet. For example, you could use a body surface area method to extrapolate to how many milligrams of Mg per body weight, the dose you gave would correspond to humans, or at least, to how many milligrams of Mg per day the dose you gave would correspond in an average human diet.

We appreciate this interesting comment. According to previous studies, Mg levels equal or around to 0.1% has been stablished in many previous studies addressing the effect of dietary Mg supplementation (e.g. Laurant et al, Br J Nutr 2000; Finckenberg et al, J Hipertens 2005; Devaux et al, Magnes Res 2016; ElZohary et al, PLos One 2016; García-Legarreta et al, Nutrients 2020). And a 0.6% magnesium (our supplementation level) was given as a dietary supplementation (high-Mg diet) in many of them and others (e.g. Laurant et al, Br J Nutr 2000; Finckenberg et al, J Hipertens 2005; ElZohary et al, PLos One 2016; Katakawa et al, Hypertens Res, 2016; García-Legarreta et al, Nutrients 2020). It must be kept in mind that if diets contain factors that might reduce the absorption of magnesium, a slightly higher dietary concentration might be required (NRC-SLAN, 1995). In this case, it is known that both a high calcium or phosphate content can reduce the absorption and/or availability of Mg and also that a decreased magnesium absorption is often found in renal disease (Brannan et al, J Clin Invest 1976). Thus, a dietary Mg supplementation of 0,6% is a reasonable amount to allow significant changes in Mg absorption after 28 days of treatment. This is also supported by the fact that no adverse effects typically related to excessive Mg intake were observed in the experimental animals.

As stated in the previous answer, the mean daily intake of Mg per body weight for animals receiving Mg 0.1% was 43.6 mg/kg. We have used the ‘dose-by-factor’ approach on the basis of body surface area to obtain the ‘human-equivalent dose’ (HED) (USFDA, 2005). Thus, this Mg intake in the rats corresponds with a HED of 7.05 mg/kg day, which assuming the human weight to be 60 kg, results in a daily intake of 420 mg that is in the range of the US recommended daily allowance (RDA).

Additionally, mean daily intake of Mg per body weight for animals receiving Mg 0.6% was 261.6 mg/kg, which then corresponds with a daily intake of 2520 mg (assuming the human weight to be 60 kg). Undoubtedly, this amount of Mg appears very high for humans. Though dosages used in clinical trials are very variable and depends on the therapeutic purpose and dosages as high as 1000 or even 2500 mg per day has been used for pain treatment and as laxative, respectively, they are ranging largely from 200 to 600 mg per day (Elgar, Nutr Med Rev. 2021). Furthermore, it must be kept in mind that these must be added to the diet. Thus, we are conscious that despite the fact that the level of our experimental dietary 0.6% Mg supplementation is acceptable for nephrectomized rats fed a high phosphate diet in the short term, it cannot be directly extrapolated to the human setting. Conversely, though our results might support a possible beneficial effect of Mg supplementation, large clinical trials of are needed to stablish a safe therapeutic windows of Mg dosage, as well as to ensure in which patients, more likely those with hypomagnesemia, supplementation might be indicated.

As a conclusion of this response to the reviewer comment, new statements with this information have been included in the revised version of the manuscript as a limitation of the study (Page 16, Line 572).

“There are limitations in this study that deserve consideration. We are aware that despite the fact that the 0.6% Mg supplementation of our experimental diet is acceptable for nephrectomized rats fed a high phosphate diet in the short term, this Mg supplementation cannot be directly extrapolated to the human setting. Conversely, though our results might support a possible beneficial effect of Mg supplementation, appropriate clinical trials would be desirable to stablish a safe therapeutic windows of Mg dosage, as well as to ensure in which patients, more likely those with hypomagnesemia, supplementation might be indicated.”

Laurant P et al. Dietary magnesium intake can affect mechanical properties of rat carotid artery. Br J Nutr. 2000 Nov;84(5):757-64

Finckenberg P et al. Magnesium supplementation prevents angiotensin II-induced myocardial damage and CTGF overexpression. J Hypertens. 2005 Feb;23(2):375-80.

Devaux S et al. Dietary magnesium intake alters age-related changes in rat adipose tissue cellularity. Magnes Res. 2016 Apr 1;29(4):175-183.

ElZohary L et al. Mg-supplementation attenuated lipogenic and oxidative/nitrosative gene expression caused by Combination Antiretroviral Therapy (cART) in HIV-1-transgenic rats. PLoS One. 2019 Jan 22;14(1):e0210107.

García-Legorreta A et al. Effect of Dietary Magnesium Content on Intestinal Microbiota of Rats. Nutrients. 2020 Sep 22;12(9):2889.

Katakawa M et al. Taurine and magnesium supplementation enhances the function of endothelial progenitor cells through antioxidation in healthy men and spontaneously hypertensive rats. Hypertens Res. 2016 Dec;39(12):848-856.

NRC-SLAN, 95-National Research Council (US) Subcommittee on Laboratory Animal Nutrition. Nutrient Requirements of Laboratory Animals: Fourth Revised Edition, 1995. Washington (DC): National Academies Press (US); 1995.

Brannan PG et al. Magnesium absorption in the human small intestine. Results in normal subjects, patients with chronic renal disease, and patients with absorptive hypercalciuria. J Clin Invest 1976;57:1412–8.

Diaz-Tocados JM et al. Dietary magnesium supplementation prevents and reverses vascular and soft tissue calcifications in uremic rats. Kidney Int. 2017;92:1084-1099.

            USFDA (2005). Guidance for Industry: Estimating the Maximum Safe Starting Dose in Adult Healthy Volunteer. US Food and Drug Administration: Rockville, MD

            Elgar K. Magnesium: A Review of Clinical Use and Efficacy. Nutr Med Rev. 2021; 1: 1-20.

  1. In the abstract, you mention that control rats have undergone a sham operation, however there is no mention of this procedure in section 2.1.1 on surgical procedures, from which it would appear as if control rats did not undergo any surgical procedure at all. You may add a sentence between lines 90 and 91, stating something like “A subset of 10 rats, that served as controls, underwent a sham-surgery without actual nephrectomy”.

Thanks a lot for this comment and the suggestion. We forgot to include this information, which has been included as the reviewer suggested.

Page 2, Line 95: “A subset of 10 rats, that served as controls, underwent a sham-surgery without nephrectomy”.

  1. Also, there is no mention in the methods section about urine collection and what analyses were performed in urines and using what methods.

Again, we apologize for the mistake. This information has been included in page 3, line 102.

“Twenty four-hour urine was collected in metabolic cages for measurement of phosphate and magnesium amount. Urine phosphate and magnesium were measured by spectrophotometry (Bio-Systems, Barcelona, Spain)”.

  1. In Line 209, in the Statistics section, you say that values are expressed as means±SE. However, when results are reported (eg in Figures 1, 2 and 4) the legend says that data are means±SEM. One of the two must be wrong.

There was a typo mistake of Statistics section. Values were expressed as mean±SEM. This has been corrected in the new version of the manuscript.

  1. In Table 1, at the line for Glucose, it would appear as if there is no statistically significant difference between the Mg0.1 and Mg0.6 groups, but there clearly is a major difference, so probably there’s a missing “###” sign.

We have reviewed the statistical analysis and have checked that the data is correct. According to ANOVA test, the difference in serum glucose levels between Mg0,1% and Mg0,6% was p<0.07 and thus, it did not achieve statistical significance.

  1. From your discussion, it would appear that the only mechanism you propose to explain the effects of Mg is its modulation of ApoB and AGT. But this is extremely reductive, considering that Mg is involved in over 300 enzymatic pathways, and there are a lot of other mechanisms that could explain its beneficial effects (for example it involvement in arterial muscle vasoconstriction, vasomotor tone, blood pressure, blood coagulation, insulin metabolism…). I would recommend to add at least a sentence about the possible mechanisms of action of Mg, and maybe reference one or two reviews that explain the many different mechanisms through which Mg may be involved in relation to CVD protection.

Thank you for the comment. Although we have tried to mention mechanisms in the discussion, suggesting a holistic effect of Mg, the fact is that if the reviewer made the above comment indicates we have to improve this message. We have revised the discussion and included additional comments highlighting the potential effects on different pathways and mechanisms whereby Mg might to develop CVD protection. The following comment has been included in page 16, line 562:

“In addition to these vascular effects, the beneficial effects of Mg supplementation and thus, the importance in the prevention of hypomagnesemia have been widely described by others. Serum Mg concentrations ranging from 1,73 mg/dL to 2,16 mg/dL are associated with a linear decrease in the risk of cardiovascular events; as well as a nonlinear inverse association with cardiovascular events (Qu et al., 2013). Similarly, pathophysiological processes such as vascular calcifications, hypertension, arrhythmias, heart failure, coagulation, endothelial dysfunction, even cardiovascular death, are possibly related to abnormal Mg homeostasis (Gröber U et al, Nutrients 2015, 7, 8199-8226). Mg deficiency accelerates the atherosclerotic process, increases thromboxane synthesis and stimulates platelet aggregation, produces oxidative stress and inflammation, and induces the synthesis of cytokines, nitric oxide and adhesion molecules (VCAM-1 and ICAM-1), all of which ends in CVD (Bo and Pisu, 2008).

Qu, X., Jin, F., Hao, Y., Li, H., Tang, T., Wang, H., et al. (2013). Magnesium and the 1069 risk of cardiovascular events: a meta-analysis of prospective cohort studies. PloS One 8, 1070 e57720. doi:10.1371/journal.pone.0057720.

Gröber U et al. Magnesium in Prevention and Therapy. Nutrients 2015, 7, 8199-8226; doi:10.3390/nu709538

Bo, S., and Pisu, E. (2008). Role of dietary magnesium in cardiovascular disease prevention, insulin sensitivity and diabetes. Curr. Opin. Lipidol. 19, 50–56. doi:10.1097/MOL.0b013e3282f33ccc.

  1. Line 113, how could you use the Friedewald formula if you did not measure triglycerides

Thank you for this important comment. We have checked this issue and noticed that we used the Friedewald formula in a wrong way. In this study plasma Triglycerides were measured, however since the Zucker rats have MetS, they all had plasma levels of Triglycerides above 150 mg/dl. Provided that the Friedewald formula produce inaccurate results when the levels of Triglycerides are elevated (Martin S et al. Friedewald-Estimated Versus Directly Measured Low-Density Lipoprotein Cholesterol and Treatment Implications. J Am Coll Cardiol. 2013 Aug, 62 (8) 732–739), we decided to remove them from the calculation and to show values of LDL according to the following formula:

LDL= Total CHOL- HDL.

Now, we understand that in these conditions it would have been better not to include the LDL values calculated in this way. Conversely, we propose to show the parameter no-HDL cholesterol that include LDL+VLDL, being this term more precise.  Certainly, it has been described that VLDL and LDL are a risk factor for coronary heart disease and atherosclerosis (Sacks FM et al. VLDL, Apolipoproteins B, CIII, and E, and Risk of Recurrent Coronary Events in the Cholesterol and Recurrent Events (CARE) Trial. Circulation. 2000; 102:1886-1892). Therefore, this parameter could be useful to demonstrated that Mg supplementation improved the lipid profile in our experimental animals. This change has been included in the new version of the manuscript (results and discussion section).

  1. Following are some minor misspells and typos:

Lines 29-30, spell out IL-1, IL-6, ET-1 and NO 

Line 35, replace “as well as improvement in” with “, and improved”

Line 45, replace “itself is being recognized as an important risk factor of CDV” with “is being recognized as an important risk factor of CDV in itself”

Line 46, replace “components” with “component”

Line 51, replace “worsens” with “worsen”

Line 55, replace “chronic kidney disease” with “CKD”

Line 66, delete “Mg”

Line 69, replace “with” with “is”

Line 69, replace “progression CKD” with “progression”

Line 80, replace “the metabolic syndrome (MetS)” with “MetS”

Line 82, open parenthesis before Barcelona, Spain

Line 82, replace “and” with “,”

Line 92, replace “metabolic syndrome” with “MetS”

Line 107, I would hope samples were stored at “–80°C”, not 80°C!

Line 115, spell out FGF23

Line 116, abbreviate nitric oxide with NO

Line 156, add beta before -actin

Line 210, “with statistical.”: something is missing, probably the name of the statistical software.

Line 212, merge sentence with line 211

Line 213, replace “was considered” with “were considered”

Line 215, spell out PCA

Line 278, delete “it was observed”

Line 292, in subfigures D and E please clarify on what tissue was the analysis performed (eg. “Nuclear extracts from aortic tissue”)

Line 339, delete the second “A)”

Line 360, replace “normalized” with “normalization”

Line 441, replace “improve” with “could improve”

Line 444, replace “there was” with “had”

Lines 455-457, spell out the acronyms

Line 468, replace “to produce GPx” with “for production of GPx”

Line 468, replace “excreted” with “secreted”

Line 504, spell out RAS

Line 506, replace “has” with “have”

Line 511, replace “other” with “another”

Line 512, replace “though” with “through”

Line 519, replace “includes” with “include”

We thank the kind support of this reviewer. All typos and misspelled have been corrected.

Reviewer 2 Report

Lopez-Baltanas R and collaborators evaluate the effect of a diet high in magnesium on oxidative stress, inflammation and vascular dysfunction in an experimental model of metabolic syndrome in zucker rats with renal failure induced by 5/6 nephrectomy. The effect of magnesium supplementation has already been evaluated by previous studies both in animal models and in clinical trials: Kidney int 2017; 92:1084-1099. Clinical and Experimental Nephrology 2022; 26: 379-384. The present study fully explores the mechanism of action of magnesium supplementation on the improvement of renal function, inflammation and oxidative metabolism. However there are some points that could improve the study: I would add a histological study to demonstrate that in addition to renal function, magnesium supplementation improves parenchymal changes; sirolimus seems to decrease serum magnesium levels (Am J Physiol Renal Physiol 2009;297:F916-22) and yet, as demonstrated by Esposito and collaborators, it improves renal alterations induced by ischemia-reperfusion injury (Am J Nephrol 2011;33:239-249) and in a 5/6 nephrectomy model (Tranpsplant Proc. 2009;41:1370-1371). In the discussion the authors should speculate on these apparently conflicting results

Round 2

Reviewer 2 Report

After carefully reading the revised version of the manuscript I think that the authors have adequately responded to the comments of the reviewers. The revised version of the manuscript can then be accepted for publication